# Video Observation of Kindergarten Teachers' Use of Questions in Picture-Book Reading with Quiet Multilingual Children: A Pilot Study

**Marit M. Bredesen [1,2] and Kari-Anne B. Næss [1,2,*]**

[1] Department of Special Needs Education, University of Oslo, Mail Box 1140 Blindern, 0318 Oslo, Norway; m.m.bredesen@isp.uio.no

[2] Department of Pedagogy, Religion and Social Studies, Western Norway University of Applied Sciences, Mail Box 7030, 5020 Bergen, Norway

[*] Correspondence: k.a.b.nass@uv.uio.no; Tel.: +47-92240741

**Abstract:** Teacher questions asked during picture-book reading may stimulate the child's practice of new vocabulary. However, there is great variation in children's amount of verbal expression, and little knowledge exists about what level of openness in the questions elicits a response. We use video observation and pilot a set of digital picture-book dialog materials that are under development. The analysis included 234 questions asked during picture-book reading in the Norwegian language between three quiet multilingual children and their kindergarten teachers. The analysis was partly qualitative evaluating the types of questions and subsequent responses and partly quantitative in summarizing the occurrence of the types of questions and responses. The results show that between 75% and 97% of the half-open questions, between 60% and 80% of the closed questions, and between 14% and 60% of the open-ended questions elicited a response from the children. Overall, the results indicated that the frequency of responses varied both within and between question types. The fact that open-ended questions generated a limited number of responses among multilingual children may challenge the use of such questions as the gold standard in adult–child dialogs, regardless of child factors and context.

**Keywords:** dialogs; question types; vocabulary; multilingual children

## 1. Introduction

Vocabulary, or knowing the meanings of words, is associated with a child's academic [1–4], social, emotional, and behavioral outcomes [5–8]. One activity that is assumed to stimulate vocabulary development is picture-book reading [9–14]. Even in simple books, there is often a higher frequency of complex words than there are in natural dialogs not supported by any written material [12]. In addition, dialogs about content give children the opportunity to learn new words in a meaningful context [15]. Questions asked during picture-book reading dialogs invite the child to reflect on and practice new vocabulary [16]. However, there is in general a great variation in children's amount of verbal expression, and to what extent they respond to questions in dialogic interactions. One group with an especially high risk of having low vocabulary [17] and silent periods [18] consists of multilingual children.

Due to increased migration over recent decades, the number of multilingual children in kindergartens has increased considerably [19]. Multilingual children are a heterogeneous group according to cultural background, time, and type of language exposure in their second language, as well as their level of language achievement [17,20].

Despite this wide variation, on average, research shows that they score significantly lower than their monolingual peers on measures of vocabulary [17,21] and therefore are in need of effective vocabulary interventions. The aim of the present paper is to pilot a

digital picture-book reading session from a vocabulary intervention under development and examine the feasibility of the questions included in the book. The results are expected to be important to the further development of this specific vocabulary intervention and to other researchers planning for picture-book interventions, and they might also be of interest to kindergarten teachers and parents who are interested in how to stimulate (their) child's language.

This paper takes both a linguistic approach, according to which types of questions receive responses from one's interlocutor, and a sociological approach, according to which the response is influenced by the social interaction between the interlocutors [22]. The main issue of interest is the level of openness in each question asked and the subsequent response. However, when needed, the questions are categorized based on the sequential initial turns. Thus, we do not ignore the fact that questions constitute part of a larger dialog sequence and are neither produced nor answered in isolation. However, it would be impossible to consider every aspect of complex social interactions, including the setting and broader social and cultural context; the interlocutors' backgrounds, experiences, and interests; paralinguistic features and elements of speech such as speed, pitch, intonation, and nonverbal communication with gesture and gaze; and various linguistic features including the words used and how they are understood, in an analysis [22]. Due to the lack of research tools that would be needed to investigate the interplay among all the relevant aspects of such a complex phenomenon, this paper focuses on a part of it: the openness of questions and their responses.

### 1.1. Picture-Book Reading and the Use of Questions

In general, questions are more frequently responded to than narrative statements such as non-question comments or prompts [23]. This is possibly a general quality of questions [23]. Questions may therefore be an effective way of eliciting verbal responses from children but may vary in both linguistic and cognitive requirements. According to Walsh and Hodge [24], a lack of common terminology across studies challenges comparisons across question types. Questions can, for example, be categorized into dichotomous categories such as closed or open-ended, eliciting or non-eliciting, contextualized or decontextualized, literal or inferential, topic initiating or topic continuing, and high- or low-cognitive-demand questions. The same question can thus have several labels and fit into different categories depending on the research focus of interest.

### 1.2. Closed and Open-Ended Questions

The classification of questions as closed vs. open-ended is widely used, and clear definitions of these two categories enhance the reliability of coding [25]. Walsh and Hodge [24], however, revealed a lack of consensus concerning the definitions of these question types in their systematic review. Examples of different descriptions of closed and open-ended questions are presented in Table 1.

As shown in Table 1, the categorization of questions as closed or open-ended may depend upon linguistic requirements, the cognitive level of abstraction, and/or the number of potentially correct responses. Different definitions can make categorization challenging. For example, the question *What does the girl feel?* can be interpreted as either a closed or open-ended question depending on the definition, the information given in the conversation, the number of possible responses, and the child's actual response. If the child responds only with *sad*, then the question could be classified as closed according to Wasik et al.'s [26] definition. Moreover, according to the definition of Strasser et al. [27], the question can be classified as open-ended unless the previous text has said something about the girl being sad so that the response no longer involves interpretation. Different definitions may be suitable for different research purposes. Lee et al. [25], De Rivera et al. [28], and Hargreaves [29] did not focus on the child's response but rather on the question itself. This emphasis on the question may possibly be appropriate when the adult reads with children who are

unresponsive—for example, children who give mainly one-word responses regardless of the type of question.

**Table 1.** Overview of different descriptions of closed and open-ended questions.

| Authors | Closed Questions | Open-Ended Questions |
|---|---|---|
| Wasik et al. [26] | Require one-word responses | Require more than one-word responses |
| Strasser et al. [27] | Not included in this study | Allow children to make predictions, interpret images and draw parallels to their own lives and experiences |
| Lee et al. [25] | One acceptable response exists, and the question constrains that response | Several different responses would be acceptable |
| De Rivera et al. [28] | Could be responded to with one or more words, the question constrained the child's response, and the answer is usually known to the adult | Could be responded to with one or more words, and the response is not predetermined by the question |
| Hargreaves [29] | Have one correct response and are often (but not always) factual | Often involve reasoning and judgment |

*1.3. Half-Open Questions*

Hargreaves [29] introduced a third category called half-open questions, which are questions that can be answered with a *yes* or *no* response. Yes/no questions are often treated as closed, and this, Hargreaves [29] suggests, is a simple and reliable solution. At the same time, some children use this opportunity and follow up with extended responses, for example, as a justification of their answer. Presented with half-open questions, a child is always free to respond with a simple *yes* or *no* and thus treat it as closed or give a more comprehensive answer and thus treat it as open-ended. The above author noted that the introduction of a separate category for this type of question solved a troublesome coding problem and, perhaps just as important, elucidated the differences in the responsiveness of the children observed. Hargreaves [29] found that verbally active children, to a greater extent, treated half-open questions as open-ended, while more passive and nonresponsive children treated them as closed.

*1.4. Present Pilot Study*

The purpose of the present pilot study is to examine kindergarten teachers' use of questions when reading picture-books with children and how different types of questions work in eliciting responses from multilingual children who use few words and few and short utterances in a one-on-one (teacher-child) classroom interaction when the Norwegian language is being used (hereafter referred to as "quiet children"). In general, multilingual children are at risk of weaker language development in terms of their second language due to less experience [17]. These quiet multilingual children, with limited verbal expression, can therefore be particularly vulnerable to weaker second-language development. Questions from adults during book reading may stimulate children's active verbal participation and offer them the opportunity to practice their second language. If certain types of questions are answered more often than others, then such knowledge can be valuable: first, to develop a general understanding of how to support these children's opportunities to participate in dialog and, second, to reveal knowledge that is useful to the development of vocabulary interventions for multilingual children.

There is no clear answer to which types of questions are best suited to stimulate dialog with multilingual children in kindergarten. In the present pilot study, we investigate the following research question:

To what types of questions do quiet children respond to a greater or lesser extent?

Here, a response refers to all forms of responses by the child, including nonverbal responses in the form of affirmative sounds or the nodding or shaking of the head.

## 2. Materials and Methods

The present pilot study is part of a larger Norwegian project approved by the Norwegian Centre for Research Data (reference number 983738). This larger project aims to develop and test the effects of a digital vocabulary intervention. The intervention consists of reading with digital picture-books and systematic activities and is per se in its piloting phase. As children do not start formal reading education in Norway before the children are six years of age, the picture-books used were wordless, but a written script on the bottom of the screen was available for the kindergarten teacher on days 1–4. The script included gradually more abstract questions across the days. The reason for choosing a digital book is the possibility of including animations that can explain the focus word in a more realistic way than is possible with a picture.

The present pilot study includes video observations of kindergarten teachers and children reading one of the books on day 4 (hereafter referred to as picture-book dialogs). Researchers have highlighted the need to pilot materials, strategies, and full interventions before eventually conducting large-scale randomized controlled trials [30,31]. The purpose of a pilot study "...should be to identify the necessity to modify questions or other procedures that do not elicit appropriate responses or enable the researchers to obtain rich data" [32] (p. 3).

### 2.1. Digital Picture-Book

The digital picture-book and the story behind the drawings were specially developed for this project. The content of the book builds on the national curriculum for kindergartens in Norway and had been evaluated by teachers and revised accordingly before being used in the present pilot study. The book consists of 11 picture pages made available on an iPad. An example of a page from the picture-book is presented in Figure 1. The story is built around the keyword *grow,* and a script with suggested questions is available at the bottom of each page of the book (there were no story scripts included in the picture-book on day 4). In total, 57 possible scripted questions were available to the kindergarten teachers.

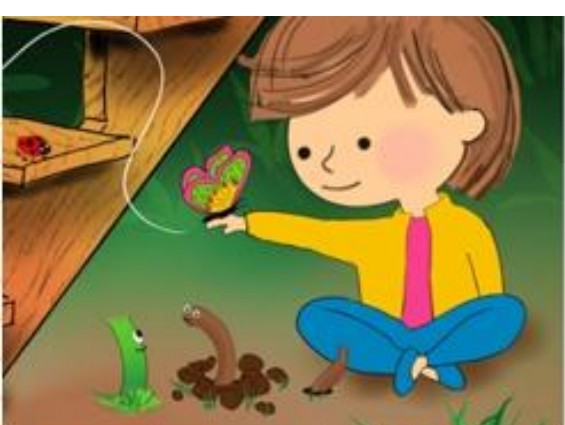

**Figure 1.** Example of a page in the picture-book (illustration Nina Skauge).

The suggested script questions regarding the picture in Figure 1 are as follows. *Can you tell me about the picture? (half-open question) What is happening here? (open-ended question) What kind of insects can you see? (closed question question) Do you know other things that grow? (half-open question) What do plants and animals need to grow? (open-ended question)*

### 2.2. Participants

The Norwegian Centre for Research Data accepted this study (registration number 983738). After obtaining ethical approval, we recruited participants from specific areas around Oslo and in the western part of Norway. Participation was voluntary, and kindergarten teachers as well as the parents or legal guardians of all the participating children

provided informed written consent. The children gave oral consent in the way that they wanted to come with the kindergarten teacher to participate in the reading session. To be included in this study, the children had to be 4–5 years old, second-language learners attending kindergarten, and have parents with a non-Scandinavian first language. We did not set any criteria regarding language proficiency in any of the child's languages or how long the child had been in Norway or attended Norwegian kindergartens. Children with diagnoses that were expected to affect language skills, such as autism and disorders of intellectual functioning, were excluded. The kindergarten teachers had to be qualified teachers. No restrictions were set in terms of teachers' experience or years since graduation.

Four kindergartens, with a total of four kindergarten teachers and 11 children, accepted the invitation to participate in the pilot study and provided signed consent.

The participants in the present study were selected from the available pilot data material with the purpose of including quiet children. The sample in this study consisted of 2 kindergarten teachers and 3 children. Kindergarten teacher 1 had worked in kindergarten for 33 years (all years in the same kindergarten) and kindergarten teacher 2 had worked in kindergarten for 24 years (all years in the same kindergarten).

All participating children had Norwegian as a second language. The categorization of verbal expression (quiet child) was conducted independently by the first author, who has a master's degree in speech and language therapy. It was performed based on video observations of the child's participation in the adult–child dialogs in the present pilot project, and it was categorized based on a relative frequency above 30% of no response to questions asked and the expression of a limited number of unique words (has an average mean length of utterance of one unique word or less). See Table 2 for other key information about the participating children.

**Table 2.** Background information about participating children.

|  | Child 1 | Child 2 | Child 3 |
|---|---|---|---|
| Gender | Female | Female | Female |
| Age | 4 years, 10 months | 4 years, 6 months | 4 years, 6 months |
| Country of birth | X | X | Undisclosed |
| Language | Kurdish, X | Kurdish, X | Serbian, English, X |
| Kindergarten [a] | 3.5 years | 3.5 years | 2 years |
| Known difficulties | None | None | None |
| Words per response [b] | 1.60 | 1.02 | 0.44 |

[a] Number of years in kindergarten; [b] Based on video observations of the child's participation in a picture-book dialog.

Kindergarten teacher 1 carried out picture-book dialogs with Child 1 and Child 2, while Kindergarten teacher 2 carried out picture-book dialogs with Child 3. As the children and the kindergarten teachers had been in the same kindergarten for years, they had been familiar with each other for a long time.

### 2.3. Data Collection

The picture-book dialog was carried out with each child separately and took place in a separate room at the kindergarten to prevent any disturbances. At the time of the data collection relevant for the present pilot study, the picture-book had already been read on three previous occasions (days). Beforehand, the kindergarten teachers had received training in how to carry out the book-reading sessions over the week. They were told that one goal of the intervention was to stimulate active verbal participation from the child and that questions (like the ones suggested in the script) may contribute to this, but at the same time, it was important to adapt the book reading session to the child in question depending on the child's initiative. The questions suggested in the script differed in regard to the level of difficulty, and the kindergarten teachers were instructed to help the children with easier (more concrete) questions if the difficult (more abstract or open) questions were hard for

them to respond to. They were also told that the goal of day 4 was for the child to be the teller of the story.

Because of the COVID-19 situation, the kindergarten teachers were given a video camera to record the picture-book dialogs. The video recordings were stored according to the General Data Protection Regulation (GDPR) on the Services for Sensitive Data (TSD) server.

### 2.4. Transcription

The recordings of the picture-book dialogs were transcribed in line with conventions, as shown in Table 3.

**Table 3.** Transcription conventions.

| Marking | Explanation |
|---|---|
| [ | Indicates overlapping talk |
| = | Indicates that the utterances follow each other without pause |
| (2) | Indicates timed pauses in seconds |
| (.) | Indicates a micro pause |
| (xxx) | Indicates indecipherable talk |
| (ord) | Indicates that the transcriber is unsure about the correctness of the transcription |
| (( )) | Indicates nonverbal expressions |
| (. . .) | Indicates that words (e.g., names) are excluded from the transcription |
| . . . | Indicates that the kindergarten teacher pauses for the child to complete a sentence or a word |

Note. The conventions are modified from Engevik et al. [33].

For a more detailed description of the transcription procedure, see Supplementary Materials S1.

### 2.5. Coding of Teachers' Questions

In this pilot study, we did not consider that the participants had read the book before. All questions were coded as if they were asked for the first time. All the questions that received responses were included in the coding process. The analysis also included unanswered questions if they were followed by a pause of three seconds or more. The cutoff of three seconds was based on Wasik and Hindman [4]. They emphasized that the child must be given enough time to respond and recommended a pause of at least three seconds before continuing the dialog. The cutoff of three seconds has also been used in a range of other studies [33]. During the transcription process, a stopwatch was used to decide whether three seconds had elapsed between the end of a question and the adult's next utterance.

The kindergarten teachers' questions were classified in terms of *type* as (1) half-open, (2) closed, or (3) open-ended. Each question was also classified by *subtype*.

Half-open questions can be answered with a 'yes', 'no', affirmative sound, nod, or shaking the head (hereafter yes-/no-type response), but they can also be answered with words other than this (there were no yes-/no-type of responses in the two other main categories). The subtypes of the half-open questions were (1) half-open closed (HO:C) and (2) half-open open (HO:O), based on the child's response.

Closed questions are defined as those wherein a particular response is expected. The subtypes of the closed questions were (1) completion, (2) localization, (3) labeling, (4) attribute, (5) questions with response options, (6) recall, or (7) closed decontextualized.

Open-ended questions are defined as those for which several different responses are acceptable. These questions usually require higher cognitive skills, such as reasoning and judgment. The subtypes of the open-ended questions were (1) summary, (2) descriptions, (3) assessment of emotions, (4) inferential, (5) open-ended decontextualized, or (6) predictions.

The main categories of all three question types were coded according to the questions themselves based on their possible responses or the types of responses they encouraged. For a more detailed description of the main categories and the different subtypes, see Supplementary Materials S1.

### 2.6. Coding of Children's Responses

The coding of the responses focused mainly on the relevance to the content rather than the linguistic sophistication. Responses to the questions were classified as:

(1) do not know/no response
(2) yes-/no-type
(3) with words
(4) with inadequate words

Criteria for the different response types were:

1. When a child did not respond or answered, 'I don't know', the response was classified as 'do not know/no response.'
2. 'Yes-/no-type' denotes those responses with yes, no, affirmative sound, nodding, or shaking of the head.
3. 'With words' denotes adequate responses with words other than 'yes-/no-type' or 'do not know.' A response was considered adequate even if it was not fully correct based on the question asked.
4. 'With inadequate words' denotes responses with words other than 'yes/no' or 'do not know', where the response was directly wrong or completely irrelevant to the question asked.

### 2.7. Data Analysis

First, after the dialog was transcribed, questions that were not related to the content of the book (for example, *Are you tired today?*) or undecipherable talk were excluded. Second, in further analysis, the questions and corresponding responses were classified. Third, frequency analysis was conducted in Excel (version 2019). Due to the sample size, no significance testing was performed in this pilot study. Therefore, any references to variation in incidence do not refer to statistically significant differences.

### 2.8. Double Coding

A total of 234 questions were coded. A trained research assistant double-coded 100% of these questions in order to ensure that the code categories we used were well described and that multiple ratings for the same question were constant. For descriptions of the double coder, see Supplementary Material S2. Thus, there were 234 codings of type and 234 codings of subtype. With regard to type, there were 7 discrepancies between the coders, and regarding subtype, there were 10 discrepancies, giving an interrater agreement (Cohen's Kappa) of 0.967 and 0.943, respectively. Coding discrepancies were resolved via discussions between the two authors until an agreement was reached.

## 3. Results

The dialogs lasted from approximately 9 min to 13 min. Child 1 was asked 92 questions (7.03 questions per minute), Child 2 was asked 71 questions (6.15 questions per minute), and Child 3 was asked 71 questions (7.61 questions per minute), which meant that Child 2, on average, received one new question (that was followed by a pause of a minimum of 3 s) approximately every 10th second. For Children 1 and 3, this number was slightly higher.

Table 4 shows that the most frequently asked question type was half-open, followed by closed, and finally, open-ended. This pattern applied to all three dialogs. Table 5 shows the relative occurrence of each response type for the half-open-ended questions.

There was a pattern across all the recorded picture-book dialogs included in the present pilot study in that the half-open questions elicited a high proportion of yes-/no-type responses and, to a lesser extent, responses of other types. This was particularly prominent for Children 2 and 3. Child 1 had the lowest proportion of yes-/no-type responses and answered more often with other words compared with Children 2 and 3. Approximately half of Child 1's responses (that were not yes-/no-type) were inadequate.

**Table 4.** Frequency of questions by type reported as a percentage.

| Type | Child 1 % | Child 2 % | Child 3 % |
|---|---|---|---|
| Half-open | 40 | 45 | 45 |
| Closed | 33 | 31 | 35 |
| Open | 27 | 24 | 20 |
| Total | 100 | 100 | 100 |

**Table 5.** Half-open questions by response type reported as a percentage.

| Response Type | Child 1 % | Child 2 % | Child 3 % |
|---|---|---|---|
| No response | 3 | 9 | 25 |
| Yes-/no-type | 67 | 91 | 72 |
| With words | 14 | 0 | 3 |
| Inadequate | 16 | 0 | 0 |
| Total | 100 | 100 | 100 |

Table 6 shows the relative occurrence of each response type for the closed questions.

**Table 6.** Closed questions by response type reported as a percentage.

| Response Type | Child 1 % | Child 2 % | Child 3 % |
|---|---|---|---|
| Do not know/No response | 20 | 45 | 40 |
| With words | 40 | 50 | 52 |
| Inadequate | 40 | 5 | 8 |
| Total | 100 | 100 | 100 |

In all three dialogs, approximately half of the closed questions were answered adequately with words (other than yes/no/I do not know). Again, Child 1 had a relatively high proportion of inadequate responses and showed the same tendency as that with the half-open questions. At the same time, Child 1 had a lower proportion of do not know/no response answers compared with the two other children. Children 2 and 3 had a relatively low occurrence of inadequate responses and, at the same time, a higher proportion of do not know/no response. Table 7 shows the total number of closed questions by subtype and the occurrence of each response type for different subtypes.

As Table 7 shows, closed questions of the subtypes *completion* and *labeling* were used relatively often in all three picture-book dialogs. The other subtypes were used infrequently and to varying degrees between dialogs.

Table 8 shows the relative occurrence of each response type for the open-ended questions.

A relatively high proportion of the open-ended questions resulted in do not know/no response. This pattern emerged across all three dialogs. In Child 1's case, 40% of these questions elicited a do not know response or no response at all. Children 2 and 3 had particularly high proportions of this response type, 65% and 86%, respectively. At the same time, Child 1 had a higher proportion of responses with words in total (adequate and inadequate) than do not know/no response, while the opposite was true for Children 2 and 3. Again, Child 1 stood out with a relatively high proportion of inadequate responses (24%), while Children 2 and 3 had no inadequate responses (0%).

Table 9 shows the total number of open-ended questions by subtype and the occurrence of each response type for different subtypes.

**Table 7.** Closed questions by subtype and response type reported as a raw score.

| Subtypes | Child 1 | | | | Child 2 | | | | Child 3 | | | |
| --- | --- | --- | --- | --- | --- | --- | --- | --- | --- | --- | --- | --- |
| | Total | D/N [a] | WW [b] | IA [c] | Total | D/N [a] | WW [b] | IA [c] | Total | D/N [a] | WW [b] | IA [c] |
| Completion | 15 | 1 | 7 | 7 | 5 | 2 | 2 | 1 | 15 | 6 | 8 | 1 |
| Location | 3 | 2 | 1 | - | 3 | - | 3 | - | 3 | 2 | 1 | - |
| Labeling | 12 | 3 | 4 | 5 | 8 | 7 | 1 | - | 4 | 1 | 3 | - |
| Attribute | 0 | - | - | - | 0 | - | - | - | 0 | - | - | - |
| Question with response options | 0 | - | - | - | 5 | 1 | 4 | - | 2 | 1 | - | 1 |
| Recall | 0 | - | - | - | 1 | - | 1 | - | 0 | - | - | - |
| Closed decontextualized | 0 | - | - | - | 0 | - | - | - | 1 | - | 1 | - |

[a] Do not know/no response. [b] With words. [c] Inadequate.

**Table 8.** Open-ended questions by response type reported as a percentage.

| Response Type | Child 1 % | Child 2 % | Child 3 % |
| --- | --- | --- | --- |
| Do not know/No response | 40 | 65 | 86 |
| With words | 36 | 35 | 14 |
| Inadequate | 24 | 0 | 0 |
| Total | 100 | 100 | 100 |

**Table 9.** Open-ended questions by subtype and response type reported as a percentage.

| Subtypes | Child 1 | | | | Child 2 | | | | Child 3 | | | |
| --- | --- | --- | --- | --- | --- | --- | --- | --- | --- | --- | --- | --- |
| | Total | D/N [a] | WW [b] | IA [c] | Total | D/N [a] | WW [b] | IA [c] | Total | D/N [a] | WW [b] | IA [c] |
| Summary | 2 | 1 | 1 | - | 0 | - | - | - | 0 | - | - | - |
| Descriptions | 12 | 5 | 6 | 1 | 10 | 8 | 2 | - | 11 | 9 | 2 | - |
| Assessment of emotions | 1 | - | 1 | - | 0 | - | - | - | 1 | 1 | - | - |
| Inferential | 7 | 4 | 1 | 2 | 1 | - | 1 | - | 1 | 1 | - | - |
| Decontextualized | 3 | - | - | 3 | 1 | - | 1 | - | 1 | 1 | - | - |
| Predictions | 0 | - | - | - | 1 | 1 | - | - | 0 | - | - | - |

[a] Do not know/no response. [b] With words. [c] Inadequate.

Of the six subtypes of open-ended questions, *descriptions* occurred most frequently in all three picture-book dialogs. The other subtypes were used infrequently and to varying degrees among dialogs. Descriptions are typically questions where the child is asked to talk about something with the support of the pictures in the book. An example of a question in this category is *What is happening here?*

Table 10 is based on the occurrences displayed in Table 9 and shows the proportion of do not know/no response vs. response for the subtype descriptions. Adequate and inadequate responses to these questions were combined to indicate their ability to elicit a response from the children.

**Table 10.** Subtype descriptions and response types reported as a percentage.

| Subtype | Child 1 | | Child 2 | | Child 3 | |
| --- | --- | --- | --- | --- | --- | --- |
| | D/N [a] | WW/IA [b] | D/N [a] | WW/IA [b] | D/N [a] | WW/IA [b] |
| Descriptions | 42 | 58 | 80 | 20 | 82 | 18 |

[a] Do not know/no response. [b] With words/inadequate.

As Table 10 shows, a relatively high proportion of the subtype *descriptions* resulted in do not know/no response in all three dialogs, especially for Children 2 (80%) and 3 (82%).

Table 11 is a compilation of Tables 5, 6 and 8 and displays the occurrence of responses to the different types of questions. All forms of responses are included. The table indicates the ability of different question types to elicit some kind of response from the children.

**Table 11.** Compilation of response occurrences from Tables 5, 6 and 8 reported as a percentage.

| Question Types | Child 1 Responses [a] % | Child 2 Responses [a] % | Child 3 Responses [a] % |
|---|---|---|---|
| Half-open | 97 | 91 | 75 |
| Closed | 80 | 55 | 60 |
| Open | 60 | 35 | 14 |

[a] Includes all forms of responses (yes-/no-type, with words and inadequate for the half-open questions and with words and inadequate for the closed and open-ended questions).

As Table 11 shows, there is a pattern across the three dialogs in that the half-open questions had a high response frequency. The closed questions had a medium-high response frequency, and the open-ended questions had a relatively low response frequency. These tendencies applied to all children, even though the response rates for different questions were not the same. Child 1, for example, responded to 60% of the open-ended questions, while Child 2 responded to 35% of the open-ended questions, and Child 3 responded to only 14% of this question type. Notably, Child 1 had a mean of 1.00 words per response to the half-open questions, 1.63 words per response to the closed questions, and 2.40 words per response to the open-ended questions. Child 2 had a mean of 0.78 words per response to the half-open questions, 0.82 words per response to the closed questions, and 0.88 words per response to the open-ended questions. Child 3 had a mean of 0.22 words per response to the half-open questions, 0.64 words per response to the closed questions, and 0.57 words per response to the open-ended questions.

## 4. Discussion

The results from the present pilot study showed that the quiet children overall were asked more questions than suggested by the scripts, possibly due to their generally low responsivity. However, the results indicated that the children responded differently to different types of questions. Between 75% and 97% of the half-open questions elicited a response from the children (mainly yes-/no-type responses). Between 60% and 80% of the closed questions elicited a response (with words), and between 14% and 60% of the open-ended questions elicited a response (with words). We discuss these findings below.

### 4.1. Half-Open Questions—Easy to Answer?

The half-open questions had a high response rate, but the responses were mainly the yes-/no-type. In line with Hargreaves [29], half-open questions were thus found to not be very suitable for getting the children to practice new vocabulary. Hargreaves [29] investigated how two groups of young school children (unresponsive and responsive) responded to different types of questions. The unresponsive group treated the half-open questions as closed, i.e., gave yes/no responses with no further elaborations or justifications, more often than the responsive group did. Yes-/no-type questions are the first type of questions that children understand [34], and they place relatively low linguistic demand on them [18]. Recommendations to minimize the use of yes-/no-type questions can be found in several studies because this question type may place the child in a passive role [35–37]. At the same time, removing the possibility of giving a yes-/no-type response by reducing the number of half-open questions may also present challenges for children, such as perceived pressure to speak in longer sentences [38] and/or higher demands in terms of prior language competence.

For some children, in certain developmental periods or in certain situations, it may therefore be possible that some well-designed half-open questions, rather than more de-

manding questions, could be appropriate. Well-designed half-open questions encourage more than a simple yes-/no-type answer without prohibiting this type of answer. In contrast, answering more demanding questions requires inferences and longer utterances. Not only is it conceivable that such questions exert less pressure because it is easier to respond to them, but situations in which a child does not meet the requirements set forth can also be avoided. The adult can further, through such questions, model language for children who are weak to express themselves at the same time as the question formulation invites children to take an active position in what is said. In addition, these questions represent an opportunity to expose children to target words by incorporating them into the question itself.

### 4.2. Closed and Open-Ended Questions—Suitable for Eliciting Responses with Words?

While closed questions had a medium-high response rate and open-ended questions had a low response rate, both types require responses with words (other than yes/no). However, closed and open-ended questions can differ in several ways, and these differences may in turn impact the children's responses. One such difference is that closed questions generally facilitate short responses [28], while open-ended questions generally encourage extended and elaborated answers. Hindman et al. [16] noted that open-ended questions often evoke more child speech in the form of longer and more complex responses. In line with this, Deshmukh et al. [39] found that Wh-style questions (many of them open-ended) boosted children's speech and recommended that teachers ask more questions of this type to elicit verbal responses at a higher level. Perhaps here lies a presumption that the verbal activity and responsiveness of the child are a result of the questioning practices of adults. Hargreaves [29] agreed with the notion that there is a correlation between questioning style and responsiveness/response length. At the same time, he questioned whether such a correlation is causal. If the correlation is causal, then the verbal activity of the children should increase when the children are given the opportunity to talk more—for example, by the adult asking open-ended questions. As the present study shows, open-ended questions do not necessarily always have such an effect. On the one hand, this finding is not in line with that of Deshmukh et al. [39]. It is, on the other hand, well in line with Hargreaves' [29] findings that open-ended questions do not automatically have this effect on verbal activity for all children. The less-responsive group of children he observed simply left many of the open-ended questions unanswered, similar to the children in the present study.

It can be argued that giving elaborate answers is more linguistically challenging than responding with one or two words. If so, weak vocabulary may hinder children's responses to open-ended questions. At the same time, as demonstrated by the children in the present study and as De Rivera et al. [28] noted, it is also possible to respond to open-ended questions with only one or a few words.

Open-ended questions may also involve the use of higher cognitive skills, such as reasoning and judgment [29]. Questions that require decontextualized thinking or reasoning are often considered more demanding for a child to answer [24]. The cognitive demand inherent in some types of open-ended questions can, therefore, also be another limiting factor with respect to response rate. In contrast, open-ended questions of the subtype descriptions (the most frequently used subtype of open-ended questions in the present study) are, as Van Kleeck et al. [40] noted, at the concrete level. According to Walsh and Hodge [24], questions at this level are usually not very cognitively demanding on the child.

Although it is possible to respond to open-ended questions of the subtype descriptions with only one or a few words, and even though they should not be very cognitively challenging for the children, they were often left unanswered in this study. One feature of these questions is that they can be relatively "wide" in the sense that they provide little guidance for how to answer and offer little support, focus, and direction to the child. It is therefore up to the child to take the initiative and decide what the answer should be about, in contrast to the situation under closed questions. Thus, it is possible that the absence of restrictions, which are intended to lead to extended responses, reduces the product



ion of language for these quiet children. It is also possible that the directiveness inherent in the closed questions, i.e., that the response to a large degree is predetermined by the question [28], helps steer the child's focus toward something specific and thereby possibly reduces the need for the child to take his or her own initiative. One of the common features of these quiet children was their low degree of initiative, as indicated by their tendency to treat the half-open questions as closed [29,41]. It cannot be ruled out that the lower requirement for their own initiative embedded in the closed questions may contribute to explaining the relatively high response rate among the children for this question type.

### 4.3. Implications

The high total number of questions, particularly the high number of unanswered questions, may indicate that the kindergarten teachers' strategy for interacting with quiet children is to fill the silence with new questions. In this regard, it is possible that the pretraining of these kindergarten teachers could offer some explanation. Even though the kindergarten teachers were instructed to adapt the book reading session to the child in question depending on the child's initiative, it may seem that asking questions, presumably with the goal of verbal activity among the children, became too dominant. It is thus possible that the pretraining program was insufficient in conveying to these teachers how to adjust the questioning strategies when the child was reluctant to talk and answer questions. Two direct implications for the further development of our interventions are that quiet multilingual children seem to be in need of other educational methods than questions alone and that the teachers are in need of better training before the intervention starts.

### 4.4. Limitations

One obvious limitation of this study is the small number of participants. It is therefore uncertain to what extent the findings would be valid in other contexts and with other quiet multilingual children. Additionally, only one day, with one specific book, was examined. It is possible that the book itself was not engaging enough and that the children's responsivity could have been different with another picture-book. The video recording may also have affected both the children and adults, and it cannot be ruled out that reading in a more natural setting would have proceeded differently. Other potential limitations include treating the question-response sequence out of the broader context and the decision to treat the questions as if they were asked for the first time when classifying them. Similar questions may have been asked by the kindergarten teachers during the book reading on previous days, which may, in turn, have led to a "closing" of the open-ended questions if the child sensed that a certain response was expected. A possible consequence of this is that the proportion of responses to the open-ended questions becomes too high. It should further be acknowledged that the children's prior semantic knowledge and interests, as well as cultural background, might have played a role in their responses. Also, the relationship between the child and the kindergarten teacher as well as the kindergarten teachers' pedagogical skills may have influenced the results. Thus, it is a limitation that no comparison group was included. It may be that less-quiet children would have answered in the same way. Karlsen et al. [42], e.g., found that a significant part of the conversation contributions from the kindergarten teachers consisted of closed questions, while the children's contributions were mainly short verbal or non-verbal responses.

### 5. Conclusions

The kindergarten teachers asked a high number of questions during the picture-book sessions, the results of which indicated that the quiet multilingual children in this study responded differently to different types of questions. The half-open questions had a high proportion of responses and elicited mainly yes-/no-type responses. The closed questions had a medium-high proportion of adequate responses and elicited only answers with words other than the yes-/no-type. The open-ended questions had a relatively low response rate and elicited answers only with words other than the yes-/no-type if they were indeed

answered. This finding implies that the degree of verbal responses to questions may rely not only on question type but also on factors such as the children's initiative and willingness to speak and/or their verbal expression ability. Thus, the types and number of questions used should be carefully adapted to each child and should be accounted for when planning dialogic interactions involving quiet multilingual children. To fill the silence with new questions seems not to be the perfect pedagogic strategy to use in picture-book reading with quiet multilingual children.

**Supplementary Materials:** The following supporting information can be downloaded at: https://www.mdpi.com/article/10.3390/educsci13101066/s1, Supplementary Information S1 Classification of questions, Supplementary Information S2 introduction to double coding.

**Author Contributions:** Conceptualization, M.M.B. and K.-A.B.N.; methodology, M.M.B. and K.-A.B.N.; formal analysis, M.M.B.; data collection, M.M.B. and K.-A.B.N.; resources, K.-A.B.N.; data curation, M.M.B. and K.-A.B.N.; writing, M.M.B. and K.-A.B.N.; project administration, K.-A.B.N.; funding acquisition, K.-A.B.N. All authors have read and agreed to the published version of the manuscript.

**Funding:** This research was funded by the Research Council of Norway, grant number 299197.

**Institutional Review Board Statement:** This study was approved by the Norwegian Center for Research Data reference number 983738.

**Informed Consent Statement:** Written informed consent has been obtained from the parents of the children to publish this paper.

**Data Availability Statement:** The data will be made available in a written format on Open Science Framework.

**Acknowledgments:** We would like to thank the participating children and parents, kindergartens and kindergarten teachers, the SL+ team and research assistants Susanne Haakonsen and Øyvind Sundstrøm. Finally, we would like to thank the funder of the study: Research Council of Norway.

**Conflicts of Interest:** The authors declare no conflict of interest.

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
