# Peer review of "Video Observation of Kindergarten Teachers’ Use of Questions in Picture-Book Reading with Quiet Multilingual Children: A Pilot Study"

_education, doi:10.3390/educsci13101066_

Round 1
Reviewer 1 Report
Thank you for this interesting article which, although pertaining to only three children over a single dialogic reading session, has some potential for providing insight into kindergartner teachers' use of questions in book reading with quiet multilingual children. I do have some concerns and suggestions for improvements:
The abstract could state that the research took place in Norway and presumably Norwegian language was used.
There are a few areas of the paper where terms need to better defined, for example on page 1, line 24, where the term 'ordinary dialog' is used - what exactly is meant by this? Likewise, the term 'talkative' children is used - how exactly is this defined? Also, on line 59 of page 2 the term 'narrative statements' is used - what is meant by this, exactly?
The paper claims that the teachers used dialogic reading - this claim needs to be better justified as there is no evidence in the paper of the teachers actually following up, prompting and conversing with children other than asking a range of questions, some of which were scripted. It doesn't really seem like dialogic reading was carried out. Furthermore, not much research about dialogic reading is reviewed except Whitchurch, who only talks about dialogue around pictures rather than dialogue around pictures and words in stories. It is not clear either whether the ebooks used were wordless picture stories or picture books. What was the nature of the ebooks and why were they used as opposed to print books? The authors say that the child had to retell the picture books - this is not the same as dialogic reading. Furthermore, the title of the article indicates that it is the teachers' use of dialogue that is the focus of the study, yet teacher dialogue is not described - only teacher questioning.
Not enough information is given about 'quiet children' and how they are defined, and exactly how they were identified for this study.
Table 2 - 4th row - what does this mean? Does it mean the number of years the child has been in kindergarten?
Methodology - Why were the questions double coded? What is the rationale for this?
Ethics - Details about ethics and consent are needed - how did the young children consent?
Discussion - The first line is unclear - I am not sure what the authors mean here.
The discussion could be enhanced by acknowledging that children may respond more enthusiastically and confidently to culturally appropriate texts that align with their prior semantic knowledge and interests - not enough information is given about the text in question and how it was deemed to be appropriate for the children in question. There may also be cultural factors at play in children responding.
Conclusion - it is claimed that teachers asked a 'vast' number of questions. I think that the word 'vast' is somewhat an exaggeration. The final sentence says that questions should be adapted to children's 'psychological mood' and 'language ability', yet these concepts are not really discussed in the paper until this final sentence so it is not appropriate to include this statement.
I hope my suggestions will be useful in improving this paper.
Reviewer 2 Report
Thank you for the opportunity to review this article, which I found to be very engaging. Some queries for consideration:
1. Against what criteria was a child deemed to be quiet? Did the SL therapist have a measure?
2. Were the books used likely to be unfamiliar to the children? If not, the child could have felt more confident to answer the questions, or some more than others depending on how familiar they were with the story.
3. What caused the children to be asked a different number of questions. Clarifying this would be useful.
4. A little more literature regarding second language learners could be useful
Reviewer 3 Report
I wonder why the importance of the teacher is not given more importance. The teacher and the children seem objectified.
Otherwise see the comment in the attached document. The article camouflages that it is a small-scale pilot. Check whether it can be processed into a research protocol, with clear addressing that it is a pilot.

Round 2
Reviewer 1 Report
Thank you for your revisions, which have allayed most of my concerns. I would like to propose a few minor changes to enhance the paper:
1) Still more detail is needed about the script for teachers - did this include a story script as well as suggested question, or just suggested questions?
2) Why are lines 142 to 143 in italics and did you mean to add something else here as it seems unfinished.
3) With regards the 'quiet children', I am still a little concerned about how they were identified and are being defined. Would it be better to say they are quiet in classroom interactions when the Norwegian language is being used? They may not be quiet in other contexts or when their L1 is being used. Also, what was their stage on L2 learning? Were they in the silent period?
4) I can't see anything on line 495 or nearby about culturally appropriate texts and how the ebook was deemed to be satisfactory in this regard.
Author Response
Thank you so much for taking the time to suggest changes to enhance the paper. I will answer each of the comments below in capital letters.
Kind regards,
Reviewer 1
Thank you for your revisions, which have allayed most of my concerns. I would like to propose a few minor changes to enhance the paper:
- Still more detail is needed about the script for teachers - did this include a story script as well as suggested question, or just suggested questions?
WE HAVE NOW MADE CLEAR THAT THE SCRIPT INCLUDED QUESTIONS, ONLY.
- Why are lines 142 to 143 in italics and did you mean to add something else here as it seems unfinished.
IN LINE WITH APA 7TH WE HAVE NOW CHANGED ITALIC TO THE USE OF “ “ AND ADDED PAGE NUMBER. .
- With regards the 'quiet children', I am still a little concerned about how they were identified and are being defined. Would it be better to say they are quiet in classroom interactions when the Norwegian language is being used? They may not be quiet in other contexts or when their L1 is being used. Also, what was their stage on L2 learning? Were they in the silent period?
WE HAVE NOW MADE CHANGES TO THE TEXT TO FOLLOW YOUR SUGGESTION.
- I can't see anything on line 495 or nearby about culturally appropriate texts and how the ebook was deemed to be satisfactory in this regard.
WE ARE SORRY, BUT WE DO NOT INTENDED TO MEASURE THE CULTURALLY APPROPRIATENESS OF TEXTS IN THE PRESENT STUDY. HOWEVER, WE CANNOT DISREGARD THAT CULTURAL FACTORS MAY HAVE INFLUENCED THE CHILD’S RESPONSES. THUS, WE THOUGHT IT WAS IMPORTANT TO MENTION THIS POSSIBLE BIAS TO THE READERS.
KIND REGARDS,
--
Author Response
Thank you so much for taking the time to suggest changes to enhance the paper. I will answer each of the comments below in capital letters.
Reviewer 3 comments on the resubmitted manuscript
Title of journal: Education Science
Manuscript number: education-2550272
Title of manuscript: Video Observation of Kindergarten Teachers’ Use of Questions in Book Reading
with Quiet Multilingual Children: One type Does not Perfectly Fit All
- Comment on the design
The article is a pilot that shows different styles of questions in use in a dialogical reading session. It is
still important to include in the title that it is a pilot.
Suggestion to the title:
Pilot of Video Observation of Kindergarten Teachers’ Use of Questions in Book Reading with Quiet
Multilingual Children: One type Does not Perfectly Fit All
CHANGES ARE MADE TO THE HEADING.
Comment on the coding classification
It is still to wonder why there is not a consistence in the four coding classifications in chapter 2.6.
Coding of Children’s Responses. Why isn’t all listed as the four classifications used; of course, not all is relevant to all the recording.
The author gave comments to their adjustment: If a question could be answered with yes/no, it
would be categorised as a half open question. Therefore, the categories of closed and open-ended
questions do not include the yes/no answer option.
I really do not understand this; a closed question, can give a yes/no answer. It is not necessary to
distinguish between the questions in the coding classification. There is this classification with four
codes, and the researcher use the one who fits.
Suggestions to 2.6:
The coding of the responses focused mainly on the relevance to the content rather than the linguistic
sophistication. Responses to half-open the questions were classified as
1) do not know/no response,
2) yes/no type,
3) with words
4) with inadequate words.
Responses to closed and open-ended questions were coded by the authors based on the available
video recordings as 1) do not know/no response, 2) with words or 3) with inadequate words.
WE HAVE FOLLOWED YOUR SUGGESTIONS AND HAVE MADE CHANGES DIRECTLY IN THE TEXT.
Last comment
The authors present their argumentation for the article to be published as an independent small scale article. However, they also insist that this article will provide recipe for reading picture book with children in kindergarten age: “as well as for teachers and parents that are interested in how to read picture book with children in kindergarten age.” Please be more humble of publishing these results as fundament for recipe for reading with children.
WE HAVE REWRITTEN THE THIS PARAGRAPH AND IT NOW STATES:
Many thanks,
Kind regards,
--